# Polycistronic Artificial microRNA-Mediated Resistance to Cucumber Green Mottle Mosaic Virus in Cucumber

**DOI:** 10.3390/ijms222212237

**Published:** 2021-11-12

**Authors:** Shuo Miao, Chaoqiong Liang, Jianqiang Li, Barbara Baker, Laixin Luo

**Affiliations:** 1College of Plant Protection, China Agricultural University, Beijing 100193, China; ms_cau@cau.edu.cn (S.M.); lijq231@cau.edu.cn (J.L.); 2Beijing Key Laboratory of Seed Disease Testing and Control, China Agricultural University, Beijing 100193, China; 3Shaanxi Academy of Forestry, Xi’an 712199, China; lcq19880305@126.com; 4Department of Plant and Microbial Biology, University of California, Berkeley, Berkeley, CA 94720, USA; 5Plant Gene Expression Center, United States Department of Agriculture-Agricultural Research Service, Albany, CA 94710, USA

**Keywords:** seed-borne virus, cucumber green mottle mosaic virus, RNA silencing suppressor, RNA interference, polycistronic artificial microRNA, *Cucumis sativus* L., antiviral resistance

## Abstract

Cucumber green mottle mosaic virus (CGMMV), as a typical seed-borne virus, causes costly and devastating diseases in the vegetable trade worldwide. Genetic sources for resistance to CGMMV in cucurbits are limited, and environmentally safe approaches for curbing the accumulation and spread of seed-transmitted viruses and cultivating completely resistant plants are needed. Here, we describe the design and application of RNA interference-based technologies, containing artificial microRNA (amiRNA) and synthetic *trans*-acting small interfering RNA (syn-tasiRNA), against conserved regions of different strains of the CGMMV genome. We used a rapid transient sensor system to identify effective anti-CGMMV amiRNAs. A virus seed transmission assay was developed, showing that the externally added polycistronic amiRNA and syn-tasiRNA can successfully block the accumulation of CGMMV in cucumber, but different virulent strains exhibited distinct influences on the expression of amiRNA due to the activity of the RNA-silencing suppressor. We also established stable transgenic cucumber plants expressing polycistronic amiRNA, which conferred disease resistance against CGMMV, and no sequence mutation was observed in CGMMV. This study demonstrates that RNA interference-based technologies can effectively prevent the occurrence and accumulation of CGMMV. The results provide a basis to establish and fine-tune approaches to prevent and treat seed-based transmission viral infections.

## 1. Introduction

Seed-based transmission of viruses via contaminated seed coats or seed embryos and the infection of germinating seedlings represents a major challenge to plant breeding for many crops, including cucurbits. Due to the lack of effective chemicals for virus disease control, many researchers have focused on understanding the interaction mechanisms between viruses and hosts in order to develop potential resistant materials or breed resistant cultivars [1,2]. As a seed-borne virus, cucumber green mottle mosaic virus (CGMMV), which is a member of the *Tobamovirus* genus, has spread worldwide via the international seed trade. CGMMV has a 6.4 kb single-stranded, positive-sense RNA genome containing four open reading frames (ORFs) [3]. Two co-terminal ORFs (ORFs 1 and 2) encode two proteins, a predominant 129 kDa protein and a 186 kDa readthrough protein, responsible for RNA replication. ORF 3 encodes a 29 kDa protein involved in viral cell-to-cell movement (movement protein (MP)), and ORF 4 encodes a 17.4 kDa coat protein (CP) [4] required for viral packaging and transmission. CGMMV primarily infects Cucurbitaceae members, causing mottling, mosaic patterns, and brown necrotic lesions on the stems, leaves, and fruits. Symptoms vary between different cucurbit crop species and cultivars of the same species [5]. Generally, the transmission rate of CGMMV in contaminated watermelon seeds is 1–10%, and transmission is greater than 12% in cucumber seeds [6,7]. Similar to many Tobamoviruses, CGMMV virions are stable, and virus particles on surfaces remain infectious for more than a year and can spread via mechanical transmission without insect vectors. When CGMMV-contaminated seeds are sown, virions from the seed coat can infect the germinating seedlings through tiny wounds that form during early growth, causing yield losses as high as 15% and >50% in cucumber and watermelon, respectively [8,9]. Therefore, effective methods to control CGMMV disease are prerequisites for cucurbits production. Currently, the most common method for virus control is through chemical and biological approaches. Sodium hypochlorite is the most effective disinfectant/virucidal chemical against Tobamoviruses. However, although disinfectants can remove CGMMV from the outer seed coat, they cannot remove virions present inside the seed [7]. Breeding cucurbits that are genetically resistant to CGMMV is hindered by the scarcity of resistance genes and instability due to temperature sensitivity [10]. Although cucumber plants can be protected from highly infectious strains via cross-protection, diseases can still emerge due to interactions with other viruses, mixed infections, or recombination [11], and effective biological measures are still needed for disease prevention and control.

RNA interference (RNAi) represents a practical approach for developing plant resistance and involves the silencing of genes via sequence-specific suppression or cleavage of complementary mRNA by small RNAs (sRNAs) [12,13]. Currently, virus-induced gene silencing (VIGS) and hairpin RNA-based silencing represent the main antiviral RNAi approaches used [14]. Despite their effectiveness, their antiviral effect is still affected by a large number of derived small interfering RNAs (siRNAs), which leads to off-target effects [15]. To reduce the occurrence of such events, second-generation RNAi strategies based on artificial sRNAs, such as artificial miRNA (amiRNA) and synthetic *trans*-acting siRNA (syn-tasiRNA)-mediated gene silencing, has been used to induce resistance to viral infection and modify several crops to obtain ideal agronomic traits [16,17,18]. Artificial sRNAs are produced in plants by expressing a functional miRNA or tasiRNA precursor containing modified miRNA/miRNA* or tasiRNA sequences, respectively. By using an overlapping PCR approach, the amiRNA precursor, which is obtained by replacing the original miRNA/miRNA* duplex region, is processed by Dicer-Like 1 (DCL1) to produce amiRNA/amiRNA* duplexes. The amiRNA strand is recruited by ARGONAUTE (AGO) proteins to form miRNA-induced silencing complexes to mediate post-transcription gene silencing or translation repression (Figure 1A) [19]. syn-tasiRNA precursors are first cleaved by a miRNA–AGO complex. Then, there is the conversion of one of the cleavage products into double-stranded RNA (dsRNA) by RNADEPENDENT RNA POLYMERASE 6 and sequential processing by DCL4 of the dsRNA into 21-nucleotide phased syn-tasiRNAs registered with the miRNA-guided cleavage site (Figure 1B) [20]. amiRNAs only recognize target sequences containing less than five mismatches, which confers high silencing specificity [21,22]. Multiple endogenous plant genes sharing a short conserved sequence can be silenced simultaneously. In addition, amiRNA-mediated viral resistance remains effective even at low temperatures [23]. syn-tasiRNAs can co-express a single precursor of several syn-tasiRNAs targeting multiple sites in a single viral RNA or different viral RNAs, inducing more effective, durable, and broad antiviral resistance. However, viruses can inhibit antiviral RNA silencing by expressing silencing suppressors, which support infection and reproduction; the mode of action of the silencing suppressor varies between different viruses [24]. The CMV-2b protein, which was the first identified silencing suppressor, can bind to double-stranded sRNA in vivo and in vitro [25,26]. The coat protein P38 in turnip crinkle viruses has been suggested to inhibit DCL4-mediated siRNA processing [27]. By comparing the SH and SH33b strains of CGMMV, it was found that a single amino acid substitution from E to G at aa position 480 in the intervening region of the 129 K protein is responsible for the impaired RNA-silencing activity (SSA) and siRNA-binding capability (SBC) of SH33b, resulting in attenuated symptoms [28]. However, it is unknown whether the accumulation of CGMMV caused by seed-based transmission in cucumber plants may mediate binding to amiRNA or affect the expression of amiRNA and, thereby, affect the antiviral effect of amiRNA. Therefore, it is vital to investigate whether an amiRNA-based system can induce resistance to CGMMV infections in cucumber plants in this study.

In the current study, ploycistronic amiRNA and syn-tasiRNA constructs were designed and expressed in *N. benthamiana* and *Cucumis sativus* to systematically analyze anti-CGMMV resistance as well as reveal the interaction between amiRNA and virus. Moreover, the polycistronic amiRNA was introduced to generate transgenic cucumber plants to evaluate its efficiency on CGMMV resistance in the next generation. Our experiments revealed that amiRNA and syn-tasiRNA could control the accumulation of seed-borne virus CGMMV. In the case of limited natural resistance resources, this approach can provide opportunities for breeding and post-disease management of crops infected with seed-borne viruses.

## 2. Results

### 2.1. Design of amiRNAs against Multiple CGMMV Strains

To confer resistance against different strains of CGMMV, amiRNAs were designed to target conserved regions of 25 CGMMV strains based on data obtained from the National Center for Biotechnology Information (NCBI) (Appendix A). The genome was screened using a 21-nucleotide window (TNNNNNNNNNNNNNNNNNNNN), and the published amiRNA selection criteria from the WMD3 output list (http://wmd3.weigelworld.org/cgi-bin/webapp.cgi, accessed on 20 March 2019) were applied, including up to two mismatches at position 1 or 15–21, absolute hybridization energy between −35 and −38 kcal/mole, and with a dG_amiR-target_/dG_perfect-match_ value >80% [29]. To reduce the unintended effects, potential amiRNAs targeting the identified target sites were investigated for potential off-target activities via WMD3 target search (http://wmd3.weigelworld.org/cgi-bin/webapp.cgi/page=TargetSearch, accessed on 21 March 2019) and cucumber (Chinese long) genome v2 database (http://www.cucurbitgenomics.org, accessed on 21 March 2019) analysis. Based on these standards, we selected six amiRNAs targeting different positions, of which three targeted the Rep genes, two targeted MPs, and one targeted the CP, which were designated amiR1-Rep, amiR2-Rep, amiR3-Rep, amiR4-MP, amiR5-MP, and amiR6-CP; amiR-GUS (GUS, β-glucuronidase) was selected as a control (Figure 2). We found no cucumber endogenous genes that functioned as potential targets when two mismatches were allowed.

Similar to our previous study, overlapping polymerase chain reaction (PCR) techniques were applied to generate amiRNA precursors by replacing the original miRNA/miRNA* duplex in ath-miR156, ath-miR164, and ath-miR171 backbones [30] (Appendix A). By using the mFold program, the computational prediction indicated that the RNA secondary structure of all amiRNA precursor constructs possessed correct folding parameters (Appendix A).

### 2.2. Expression and Anti-CGMMV Activity of amiRNAs in Nicotiana Benthamiana

In previous studies, we found that amiRNA expression and antiviral activity were correlated [30]. Therefore, we aimed to check whether the expression levels of the anti-CGMMV amiRNA correlated with the antiviral activity using transient expression assays in *N. benthamiana*. Here, we agroinfiltrated each construct into the plants to compare the expression of each amiRNA; –amiR-GUS was used as a control. Northern blot analysis of RNA preparations obtained three days post-agroinfiltration revealed that all amiRNAs were expressed. Among them, amiR4-MP had the highest expression level, while the expression level of amiR1-Rep and amiR5-MP were similar with the control; quantitative reverse transcription-PCR (qRT-PCR) results also confirmed that the expression levels of amiR4-MP and amiR6-CP were approximately 20- and 10-fold higher than those of the control, respectively (Figure 3A). The trend of the amiRNA expression level in CGMMV-infected *N. benthamiana* plants at 3, 10, and 15 days post-inoculation (dpi) was analyzed using a previously described assay [30]. In general, the expression level of amiRNA showed a downward trend, and the decrease in expression level slowed down after 10 dpi and was maintained at a certain level (Figure 3B).

To compare the antiviral activity of different amiRNAs, six agroinfiltrated leaves from independent plants were inoculated with CGMMV after 3 days of amiRNA treatment. We monitored the appearance of characteristic CGMMV-induced symptoms in the inoculated tissues (necrotic lesions) and upper non-inoculated tissues (deformed leaves and mottle) and detected the protein accumulation of CGMMV in upper non-inoculated leaves at 20 dpi by Western blot. It was shown that the upper non-inoculated tissues in all plants infiltrated with amiR-GUS displayed strong leaf green mottle. Plants agroinfiltrated with amiR4-MP and amiR6-CP did not show visible symptoms (Figure 3C) and displayed reduced levels of CGMMV accumulation (Figure 3D). However, plants infiltrated with amiR1-Rep and amiR5-MP showed multiple green mottles on the upper leaves at 20 dpi. The accumulation of viruses in the upper leaves of the amiR1-Rep-infiltrated plants was significantly higher than that in amiR5-MP-infiltrated lines (Figure 3C,D).

### 2.3. Identify the Most Effective amiRNA for Generating Polycistronic Constructs

To confirm the antiviral activity of each amiRNA more rapidly and directly, we generated a transient in vivo sensor system based on a previous study [23]. By introducing the target sites of amiRNA into the 3′ end of the green fluorescent protein (GFP) reporter gene, we prepared sensor constructs, designated pGFPamiR1/4, pGFPamiR2/5, and pGFPamiR3/6, containing the 21-nucleotide target sequences in the CGMMV genome for amiR1-Rep–amiR6-CP, respectively (Figure 4A). *Agrobacterium*-mediated co-transformation experiments were performed by co-infiltrating the GFP sensor constructs with the particular amiRNAs.

Based on the fluorescence intensity, it was found that amiR2-Rep, amiR3-Rep, amiR4-MP, and amiR6-CP efficiently inhibited the activity of their target regions. Among them, amiR3-Rep, which was expressed at a 3-fold higher level than those of the control in *N. benthamiana* at 3 dpa (Figure 3A), almost completely inhibited the expression of the GFP sensor construct. In contrast, amiR1-Rep and amiR5-MP, which showed a similar expression level to the control in *N. benthamiana* did not significantly inhibit the expression of the GFP sensor construct (Figure 4B). To evaluate the specificity of each amiRNA to its target, we set up a mismatch experiment group containing the anti-CGMMV amiRNA and GFP sensor constructs. By co-infiltrating the amiR1-Rep with pGFPamiR2/5, amiR2-Rep with pGFPamiR3/6, amiR3-Rep with pGFPamiR1/4, amiR4-MP with pGFPamiR2/5, amiR5-MP with pGFPamiR3/6, and amiR6-CP with pGFPamiR1/4, respectively, it was found that there is no difference in the fluorescence intensity between the mismatched group and the blank control (Appendix A). The potential biological activity of the amiRNA candidates suggested that their ability to silence the virus can be identified using GFP sensors and the amiRNA-mediated resistance effect detected in *N. benthamiana* to comprehensively compare the effect of amiRNA and not only judged according to the expression level of amiRNA.

Based on the performance of amiRNA in silencing the virus, we selected amiR2-Rep, amiR4-MP, and amiR6-CP to generate polycistronic constructs, which generated multiple amiRNAs from a single transcript and were inspired by the polycistronic miRNAs found in nature [31]. The mFold program was used to predict the secondary RNA structure of the resulting amiRNA constructs, revealing that the individual amiRNA backbones could fold correctly (Figure 5A). To test the biological activity of the polycistronic amiRNA construct, we infiltrated it into *N. benthamiana* and inoculated it with CGMMV. The expression level of amiRNAs in the polycistronic constructs was lower than that of amiRNAs alone at the same period (Figure 5B). However, there were no symptoms of CGMMV in the upper leaves at 20 dpi (Figure 5C).

### 2.4. Expression and Anti-CGMMV Activity of Polycistronic amiRNA Constructs in Protoplasts of Cucumber Infected with CGMMV

To determine whether the polycistronic amiRNA construct (amiR246) can decrease the transmission and pathogenicity of CGMMV transmitted by seeds or whether the resistance effect of amiR246 would be affected by virus accumulation, we evaluated the effectiveness of amiR246 in protoplasts of CGMMV-infected cucumber. Firstly, to simulate the natural presence of viruses on seeds, we developed a new method with reference to previous studies to prepare cucumber seeds harboring CGMMV [32] (Figure 6A). The germinated cucumber seeds were immersed in 10 mL portions of agroinfiltration solution with the *A. tumefaciens* culture mixtures containing CGMMV infectious clone vector (pCGMMV), followed by infiltration using the vacuum infiltration method and cultivation in the same agroinfiltration mixtures for 15 h, then sown. Two cotyledons were collected after they unfolded to prepare protoplasts [33], and the accumulation of CGMMV was determined by detecting CP expression. amiRNA at two plasmid contents (15 and 20 µg) was incubated with protoplasts (amiR-GUS was co-transfected as a control), and the expression of amiRNAs at three time points (18, 24, 36 h) and its influence on virus accumulation were detected. We observed a reduction in CGMMV protein levels at 18 h after co-transfection with amiR246. The addition of 20 µg of amiRNA246 resulted in approximately 44% protein silencing by 36 h, while 15 µg amiRNA246 caused approximately 39% protein silencing (Figure 6C). It was also found that amiR246 expression decreased significantly at 36 h relative to 24 h (Figure 6B). We speculated that this might be due to the influence of the CGMMV because the 480th amino acid of Rep would inhibit siRNA activity [28]. However, the effect on miRNA is unclear.

Based on the findings of a previous study [28], we mutated the 480th amino acid affecting RNA silencing suppression from glutamic acid (E) to glycine (G) in the infectious clone to test their influence on the antiviral effect of amiRNAs (Appendix A). Compared to the effect of CGMMV infection, the expression of amiRNAs did not decrease under infection with the attenuated strain CGMMV^E480G^ after 36 h, not different from that at 24 h (Figure 6B). Regarding viral protein expression, at 24 h, when the amiRNA content reached 20 µg, the inhibition rate of amiRNA on the expression of CGMMV^E480G^ reached 0.6 and the inhibition rate on the expression of CGMMV reached 0.4 (Inhibition Rate = (control group − 20 µg group)/control group). At 36 h, the accumulation of the two CGMMV strains showed significant differences under the influence of different amiRNA contents (Figure 6D).

To determine the specificity of amiR246 to the viral target region, the target sequences were mutated synonymously (CGMMV Res.) based on the characteristics of amiRNAs, recognizing a target sequence with less than five mismatches (Appendix A). Since the mutation regions were located in the conserved region, the accumulation of the virus was significantly reduced. We found that compared to the control, 15 µg amiRNA246 did not significantly affect virus accumulation after incubation for different times (Figure 6C).

To compare the effects of two artificial sRNAs against CGMMV, three 21-nucleotide mature regions were combined and inserted into the Arabidopsis *TAS1c* gene based on previous studies [20] and expressed miR173 for cleaving the *TAS1c* gene to produce artificial siRNAs. The result suggests that there is no difference in the suppression effect of two artificial sRNA on virus accumulation (Figure 6E).

### 2.5. Evaluation of Transgenic Cucumber Lines’ Resistance Following CGMMV Infection

To analyze the stability of antiviral activity of amiRNAs against CGMMV in a natural host, we developed stable transgenic cucumber lines harboring the polycistronic construct. In contrast to the previous construct, we added a GFP tag downstream of the polycistronic amiRNA construct to rapidly identify the positive transgenic lines (amiR-CGMMV) [34]. A total of two T_0_ transgenic cucumber plants were obtained (T_0_-10 and T_0_-17) (Appendix A), and genomic PCR screening for the presence of polycistronic amiRNA constructs was performed using primers LF2096 and LF2097 (Appendix A), which span from within the promoter region, the amiRNA gene, the GFP sequence, and the nos terminator.

Since T_0_ transgenic plants do not differ in morphology from wild-type plants, the polycistronic amiRNA construct did not influence plant growth or development (Figure 7A). Subsequently, T_0_ plants were self-crossed to produce T_1_ transgenic plants (T_1_-10 and T_1_-17). Northern blot analysis of RNA preparations obtained from apical leaves revealed that the polycistronic amiRNA accumulation was highly variable in different lines. Among them, amiR-CGMMV accumulation was similar in 7 of the 11 lines, whereas the lines amiR-CGMMV-10-1, amiR-CGMMV-10-8, and amiR-CGMMV-10-11 accumulated considerably higher levels of amiR-CGMMV, and the expression level of amiR4-MP increased 3- to 5-fold (Figure 7B). To test the resistance level of these transgenic plants to the virus, the T_1_ generation was subsequently infected with CGMMV via mechanical inoculation at the first true-leaf stage. Both the appearance and progression of viral symptoms at 20 dpi were observed, and CGMMV accumulation was assessed via Western blot at two different time points (20 and 40 dpi). Plants with high expression levels of amiR-CGMMV (amiR-CGMMV-10-11) were seemingly free of viral infection symptoms at all data points and indistinguishable from uninfected wild-type plants (Figure 7A). However, most amiR-CGMMV-carrying individuals were not completely resistant and exhibited a slight mottle symptom on the top leaves at 20 dpi (amiR-CGMMV-10-14, amiR-CGMMV-10-16, amiR-CGMMV-10-17) (Figure 7C). Notably, some plants showed relatively reduced viral accumulation at 40 dpi compared to 20 dpi, such as amiR-CGMMV-10-10, amiR-CGMMV-10-16, and amiR-CGMMV-10-17 (Figure 7E). At the same time, we found certain differences in height for plants with different susceptibility levels, showing that plant height is inversely related to virus accumulation (Figure 7D).

To detect how amiRNA inhibits viral expression, we compared the results of qRT-PCR and Western blot detecting the CGMMV accumulation of different lines. We found that line amiR-CGMMV-10-7 did not accumulate abnormally high levels of CGMMV RNA. For the same moderately resistant plants, the expression of CGMMV RNA in amiR-CGMMV-10-17 was not different from that in amiR-CGMMV-10-14; however, the protein accumulation was significantly lower than that of amiR-CGMMV-10-14 (Figure 7C). These results suggest that amiR-CGMMV acts on CGMMV RNAs via endonucleolytic cleavage or induces translational repression.

According to the accumulation of viral protein and the symptom of apical leaves, we divided the two transgenic families into three parts: The transgenic (signified by +) and resistant (signified by R) segregants were designated as amiR-CGMMV+R; some plants that showed intermediate height and a lower virus titer compared to susceptible control were designated as moderately resistant or amiR-CGMMV+MR; the fully susceptible amiR-CGMMV-carrying plants were designated as amiR-CGMMV+S. In the transgenic T_0_-10 family, three lines with higher amiR-CGMMV accumulation were symptom- and virus-free at two data points (amiR-CGMMV-10-1, amiR-CGMMV-10-8, and amiR-CGMMV-10-11). amiR-CGMMV-10-7, amiR-CGMMV-10-9, amiR-CGMMV-10-10, amiR-CGMMV-10-14, amiR-CGMMV-10-16 and amiR-CGMMV-10-17 conferred moderate resistance to CGMMV. We could not identify completely resistant plants in another T_1_ family (T_1_-17) (Figure 7F).

To determine whether moderately resistant or susceptible plants have weakened resistance due to mutations in the virus target sites, TS (target sites) sequences from the viral progenies of CGMMV-infected transgenic plants were analyzed via RT-PCR, followed by Sanger sequencing (Appendix A). However, no mutations were detected, possibly because the target sequences are located in the conserved region of the virus, suggesting that the difference in resistance between different plants may be unrelated to mutations in viral genomes.

## 3. Discussion

Seed-borne viruses, such as CGMMV, can be transmitted via seeds and are responsible for high yield losses for cucurbit crops [5]. However, no effective approaches are available for curing virus-infected plants. Compared to several methods, the best approach involves the cultivation of virus-resistant cultivars. According to our previous study, transient expression of amiRNAs in *N. benthamiana* can reduce virus accumulation [30]. Here, we aim to develop a polycistronic amiRNA expressing the best-performing amiRNAs against CGMMV. We detected the effect of amiRNAs in suppressing the infections of CGMMV transmitted by seeds and the degree of virus influence on amiRNA expression. We systematically analyzed the anti-CGMMV resistance induced by the expression of a polycistronic amiRNA construct in transgenic cucumber plants. The results indicated that the polycistronic amiRNA construct could effectively inhibit the accumulation of CGMMV transmitted by seeds and would simultaneously be affected by the activity of the virus-derived silencing suppressors (Figure 6B,C). The polycistronic amiRNA construct conferred genetic resistance to CGMMV in transgenic cucumber plants and excluded the factors that reduce resistance due to the mutation of the virus under amiRNA pressure.

Regardless of whether amiRNAs are transiently expressed in *N. benthamiana* or cucumber protoplasts or transgenic plants stably express amiRNAs, six amiRNAs had different antiviral effects. Even if the same amiRNAs were expressed in different transgenic lines, the viral accumulation was different. This may be due to the following reasons:

Firstly, it is generally accepted that there is a positive correlation between the accumulation of antiviral amiRNAs and the degree of induced resistance [16,35,36]. It was reported that there is a threshold level of amiRNA accumulation, below which the virus-targeting activity of amiR-TSWV is inefficient and cannot impede viral replication and spread [20]. In this study, amiR2-Rep, amiR4-MP, and amiR6-CP, which were relatively effective in silencing viral RNA, also showed high expression, whereas amiR1-Rep, which was almost undetectable in a Northern blot, did not induce resistance to CGMMV (Figure 3). From the perspective of amiRNAs, the miRNA precursor backbone, the complementarity of the amiRNA target sequence, and the free energy of the amiRNA precursor (stem-loop stability) may represent factors that affect the expression of amiRNAs [37,38,39]. According to Zhang [40], amiRNAs with perfect complementarity to their targets may require significantly less hybridization energy to anneal with their targets, which makes amiRNA more effective. Therefore, one of the selection criteria for all amiRNAs in our study was hybridization energy between −35 and −38 kcal/mole, which ensured the stable complementation between amiRNA and its target gene. Furthermore, the silencing effect could be enhanced by constructing a polycistronic structure targeting multiple sites. However, amiRNA candidates should be evaluated experimentally (for example, using the ETPamir assay or a rapid transient sensor system) to confirm their activity in plant cells or to identify the most potent candidate because the target accessibility of a given amiRNA in a cellular context is a prerequisite for efficient gene silencing, but its influence is unpredictable.

Secondly, for seed-borne viruses, the accumulation of viruses in plants affects the function of exogenously introduced amiRNAs. The silencing suppressor 2b of CMV can not only bind miRNA but also bind to the AGO1 protein in the miRNA synthesis pathway, thereby inhibiting the cleavage activity of the AGO1 protein [41]. In another seed-borne virus, the zucchini yellow mosaic virus (ZYMV), which often infects Cucurbitaceae, although its silencing suppressor HC-Pro can inhibit siRNA, it cannot suppress the binding of miRNA to AGO1 protein [42]. The CGMMV-SH strain has been found to show a stronger SBC than that of the attenuated strain SH33b [28]. In our study, we developed a new method to infect cucumber seeds using the infectious clone of CGMMV and successfully replicated and expressed CGMMV in cucumber protoplasts. We determined that the mutation of the 480th amino acid of CGMMV, which affects RNA silencing suppressor activity, can influence the antiviral effect of amiRNAs by inhibiting the expression of 21 nucleotide amiRNAs (Figure 6B). Moreover, when the dose of amiRNAs reaches a certain level, amiRNAs can overcome the antagonistic effect of the silencing suppressor and induce resistance to the virus (Figure 6C,D). In theory, amiRNAs are effective for the prevention and treatment of target seed-borne viruses and can be used in the exogenous addition of amiRNAs to control viral diseases.

Notably, similar to siRNA [43], several amiRNAs mediate potential gene silencing via translational repression and/or mRNA cleavage, leading to the decrease of virus accumulation. According to Zhang [38], certain amiRNAs mediate effective gene silencing, mainly via translation inhibition rather than mRNA attenuation, and the common practice of quantifying target mRNA levels as an amiRNA-mediated gene silencing indicator may underestimate the true level of gene silencing. Therefore, we also performed Western blotting to detect viral protein expression to evaluate amiRNA-mediated resistance and found that transgenic line amiR-CGMMV-10-17 and line amiR-CGMMV-10-14 accumulated a similar amount of CGMMV RNA, but the former showed relatively lower protein expression (Figure 7C).

Furthermore, mutant viruses may escape amiRNA surveillance [44,45]. According to previous studies, subinhibitory accumulation of amiRNA allows the viral evasion of antiviral resistance via accumulation of TS mutations, and all susceptible lines have been shown to accumulate moderate levels of amiR-TSWV compared to the resistant amiRNA lines [20]. We addressed this issue in our experiments by selecting amiRNA targets based on conserved regions in 25 full CGMMV genome sequences available; deep sequencing of virus populations from infected transgenic plants confirmed no mutations in the target sequences (Appendix A). This highlights the importance of multiple targets in polycistronic amiRNA and the importance of aligning as many virus genomes as possible to select highly conserved regions.

In the case of limited natural resistance resources and where the chemical treatment cannot completely remove virions present inside the seed, artificial sRNAs-based approaches can be used as a strategy to induce resistance to viral diseases because of their specificity and flexibility. In recent years, several bacterial CRISPR/Cas systems have been used to induce antiviral resistance [46]; a major risk of this approach involves the possible generation of virus variants, which can be avoided by designing artificial sRNAs targeting multiply conserved regions of the virus. In this study, we have demonstrated that the polycistronic amiRNA construct, expressing high-efficiency amiRNAs targeting multiply conserved regions of the virus, conferred a long-lasting resistance to CGMMV in cucumber. By simulating a scenario of seed transmission in nature, we have demonstrated that the subsequent addition of artificial sRNAs inhibited the accumulation of viruses transmitted by cucumber seeds and the activity of virus-derived silencing suppressors affected the expression of amiRNAs, which explained that amiRNAs could decrease disease severity by spraying them onto plants infected by viruses [47]. In theory, our system is suitable for the targeting of different seed-borne viruses and can be used for preventing viral diseases in cucurbit crop species.

## 4. Materials and Methods

### 4.1. Plant Materials and Growth Conditions

*N.benthamiana* and *Cucumis sativus* L. xintaimici plants were cultivated in a greenhouse at 25 °C, exposed to a 16 h light/8 h dark photoperiod. An *N. benthamiana* reference Nb-1 genotype was obtained from the Boyce Thompson Institute and used for *Agrobacterium* infiltration experiments. Seeds of *Cucumis sativus* L. xintaimici were inbred line seeds obtained from Xizhang village, Xintai City, China.

### 4.2. Plasmid Construction

amiR-CGMMV was constructed, as described in a previous study [30]. CGMMV-specific amiRNAs showing absolute hybridization energy between −35 and −38 kcal/mole were selected; the positions of amiRNA target sequences in the conserved regions of the CGMMV genome are shown in Figure 1. amiRNAs targeting the *GUS* gene were used as a control. Three *Arabidopsis thaliana* precursor miRNA backbones (athmiR156, ath-miR164, and ath-miR171) were used to generate amiRNA precursors. This strategy involved the use of overlapping PCR primers, as described by Li [29] (Appendix A). Secondary structures of the designed pre-amiRNAs were predicted using the mFold web server (http://mfold.rna.albany.edu/?q=mfold/RNA-Folding-Form, accessed on 25 March 2019). The amiRNA precursor sequences obtained from overlapping PCR were cloned into pENTR^™^/D-TOPO^®^ (Invitrogen, Waltham, MA, USA), and the clones were confirmed by sequencing. Then, the confirmed sequences were recombined into the binary expression vector pEarlyGate100 (pEG100) by the LR reaction. Expression of all pre-amiRNAs was driven by the cauliflower mosaic virus (CaMV) 35S promoter and terminated with the OCS terminator. The resulting recombinant binary expression plasmids containing the amiRNA precursor were designated as pEG100.aMIR1, pEG100.aMIR2, pEG100.aMIR3, pEG100.aMIR4, pEG100.aMIR5, and pEG100.aMIR6. The vector pEarlyGate100 (pEG100) was obtained from Barbara Baker’s Lab.

To generate the three amiRNA GFP sensor constructs, 42 base-pair-long sequences were artificially synthesized (GeneArt^®^, Life Technologies, Shanghai, China), containing the recognition sites of either amiRNAs 1/4, 2/5, or 3/6. Then, target sequences were ligated to the 3′ untranslated region of the mGFP5 reporter gene in the pEG100 vector and digested with XbaI and XhoI, resulting in the sensor constructs of pGFPamiR1/4, pGFPamiR2/5, and pGFPamiR3/6.

Three selected amiRNAs (amiR2-Rep, amiR4-MP, and amiR6-CP) were inserted into the pEG100 binary construct using a ClonExpress^®^ Cloning Kit (C112, Vazyme, Nanjing, China), thereby creating pEG100-amiR246. For cucumber transformation, pEG100-amiR246 was ligated with a GFP gene for screening amiR-CGMMV transgenic lines.

The site mutation of the CGMMV infectious clone vector was used (Mut Express^®^ MultiS Fast Mutagenesis Kit V2; Vazyme, Nanjing, China). We mutated the 480th amino acid of Rep from glutamic acid (gaa, E) to glycine (gga, G) to obtain the pCGMMV^E480G^ vector and mutated synonymously five target sites of each amiRNA in a CGMMV infectious clone to obtain the pCGMMV Res. vector.

The Arabidopsis *TAS1c* gene containing the target sequence of ath-miR173 was inserted in three mature sequences (amiR2, amiR4, amiR6). The precursors of ath-miR173 and the Arabidopsis *TAS1c* gene were inserted into the pMDC32B binary vector using a ClonExpress^®^ Cloning Kit (C112, Vazyme, Nanjing, China) (syn-tasiR246). We constructed syn-tasiR-GUS as a control.

### 4.3. Agrobacterium Tumefaciens Infiltration and Viral Infection Assays

*A. tumefaciens* GV3101 (AC1001, WeiDi, Shanghai, China) was transformed with the binary expression vectors containing amiRNAs. GV3101 carrying the pre-amiRNA expression vectors was infiltrated into *N. benthamiana* (two leaves per plant), as described previously [30]. Briefly, GV3101 was cultured till an optical density of 1.0 at 600 nm (OD_600_) was obtained, diluted to an OD_600nm_ of 0.2, and infiltrated into young leaves of *N. benthamiana* (plant age, 5–6 weeks). Viral infection assays using CGMMV No.2 isolate were performed as described previously [30].

### 4.4. Vacuum Agroinfiltration and Co-Cultivation

Cucumber seeds were surface-sterilized via soaking in 75% (*v/v*) ethanol for 20 s and rinsed five times with sterile deionized water. Then, they were soaked in 2.5% sodium hypochlorite for 6 min and rinsed five times with sterile deionized water. The sterilized cucumber seeds were placed on 2–3 layers of filter paper soaked in distilled water and germinated in an incubator set at 30 °C for 30 h in the dark. We waited for the emerging sprouts to grow to approximately 3 mm long before proceeding to the next step.

*A. tumefaciens* strains containing pCGMMV, pCGMMV^E480G^, pCGMMV Res., respectively, were grown overnight at 28 °C in Luria-Bertani (LB) medium supplemented with 20 mg L^−1^ of rifampicin and 50 mg L^−1^ of kanamycin. Subsequently, each overnight culture was centrifuged at 4000 rpm at 10 min and supplemented with resuspended liquid (acetosyringone (AS) (19.62 mg L^−1^), cysteine (Cys) (400 mg L^−1^), and Tween 20 (5 mL L^−1^)) to reach an optical density (OD_600_) of 0.3. The sterilized (germinated) cucumber seeds and *Agrobacterium* were added to a 20 mL syringe, and the plug and plunger were pulled fully to the top of the syringe, creating a vacuum of approximately 20 kPa for 30 s. Plugs were removed, and the process was repeated twice. Infected seeds and *Agrobacterium* were co-cultivated for 15 h and transferred to soil to grow for 10 days.

### 4.5. Transient Expression in Cucumber Mesophyll Protoplasts

Cucumber mesophyll protoplasts were prepared and transformed as previously described [48]. The middle region of the cotyledons from 7 to 15-day-old cucumber seedlings were collected and cut into 0.5 to 1 mm strips, digested in an enzyme solution containing 20 mM MES (pH 5.7), 1.5% cellulase R10 (Yakult Honsha, Tokyo, Japan), 0.3% macerozyme R10 (Yakult Honsha), 0.3 M mannitol, 20 mM KCl, 10 mM CaCl_2_, and 0.1% bovine serum albumin for 6–8 h, with gentle shaking at approximately 40 rpm in the dark. An equal volume of W5 buffer (2 mM MES (pH 5.7), 115.5 mM NaCl, 62.5 mM CaCl_2_, 3.75 mM KCl) was added to the enzyme/protoplast solution, and the suspension was then filtered through a 75-mm nylon filter (Solarbio Supply Company, Beijing, China). After centrifugation at 150× *g* for 2 min at 4 °C, the protoplasts were washed twice with W5 buffer and then placed on ice for 30 min. The protoplasts were resuspended in an appropriate MMg solution (2 × 10^5^/mL) (4 mM MES (pH 5.7), 0.3 M mannitol, 15 mM MgCl_2_).

For transformation, 20 µg of high-purity plasmid that was extracted using a Plasmid Maxprep Kit (DP117; Tiangen, Beijing, China) was added to 100 µL of protoplast suspension. Subsequently, 120 µL of 40% polyethylene glycol (PEG) solution (40% PEG 4000, 0.15 M mannitol, 100 mM CaCl_2_) was added, and the sample was gently mixed before incubation at room temperature (25–28 °C) for 15 min. The reaction was stopped by adding 600 µL of W5 buffer, followed by thorough mixing of the sample, centrifugation at 150× *g* for 2 min, and then washing the cells with 600 µL of W5 buffer to remove the PEG solution. Finally, 500 µL of W1 buffer was added to the cells, and the sample was incubated for 18–36 h.

### 4.6. Generation of Transgenic Cucumber Plants

The expression vector pEG100, which contained the polycistronic amiRNA construct, was transformed into *A. tumefaciens* strain GV3101 using the thermal excitation method.

The vector was transformed into the cucumber cultivar Xintaimici using the cotyledon transformation method [34]. First, the outer seed coat was removed and the cotyledon was cut into two parts after two days. *Agrobacterium* culture (*A. tumefaciens* cells harboring pEG100-amiR246) at an OD_600_ = 0.2 was used to infiltrate the seeds in a vacuum. The seeds were placed on an inoculation agar medium (MS medium supplemented with 2 mg L^−1^ 6-BA, 1 mg L^−1^ ABA, pH 5.7–5.8). After culturing in the dark at 28 °C for 3 d, the seeds were transferred to shoot regeneration medium (polarized medium supplemented with 100 mg L^−1^ kanamycin and 200 mg L^−1^ timentin). Approximately 2–3 weeks later, the GFP-positive plants were screened via fluorescence microscopy and transferred to rooting medium (MS medium supplemented with 100 mg L^−1^ timentin) for inducing root formation. After 20–30 d, when new roots were formed in the rooting medium, the seedlings were exposed to weak light for 3–4 d and then transferred to pots containing vermiculite. The transgenic plants were cultivated in an artificial climate incubator at 25 °C (day) and 18 °C (night).

### 4.7. sRNA Gel Northern Blot Assays

Total RNA was extracted from *C. sativus* L. and *N. benthamiana* leaves using TRIzol reagent (Thermo Fisher Scientific, Waltham, MA, USA) according to the manufacturer’s protocol. For Northern blot, 30 μg of total RNA was heat-treated in formamide buffer and loaded on a 12% denaturing urea-polyacrylamide gel electrophoresis gel. Subsequently, RNA samples were transferred to a Hybond-N^+^ membrane (GE Healthcare Life Sciences, Buckinghamshire, UK) and hybridized using probes for each of the six amiRNAs (Appendix A). Hybridization was performed using a standard protocol [49]. The probes were end-labeled using digoxigenin (Huada, Beijing, China) according to the manufacturer’s protocol. The membranes were incubated overnight at 42 °C, and immunoblotting was performed using an anti-digoxigenin–AP conjugate antibody (Anti-Digoxigenin-AP, Roche, Basel, Switzerland).

### 4.8. qRT-PCR

qRT-PCR was performed to measure amiRNA expression levels using a miScript SYBR^®^ Green PCR Kit (Q221, Vazyme, Nanjing, China) on an ABI 7500 fast instrument (Applied Biosystems Inc., Waltham, MA, USA) with the designed amiRNA primers (Appendix A). A 25 μL reaction mix was prepared using 12.5 μL of 2× QuantiTect SYBR Green PCR Master Mix, 2.5 μL of 10× miScript Universal primer, 2.5 μL of 10× forward primers (amiRNA primers), 6.5 μL of nuclease-free water, and 1 μL of cDNA product. The thermal cycler was set to the following conditions: initial activation step at 95 °C for 15 min, followed by 40 cycles of 94 °C for 15 s, 55 °C for 30 s, and 72 °C for 30 s. The relative expression levels of the CP genes of CGMMV were determined using specific primers (Appendix A). The mean quantification cycle value was used for calculations using the 2^−^^ΔΔCt^ method.

### 4.9. Western Blot Assays

Briefly, 1 µg of plant tissue was mixed with protein extraction buffer, vortexed for 3 min, centrifuged at 10,000× *g* at 4 °C, and then the supernatant was mixed with sodium dodecyl sulfate (SDS) loading buffer in a 3:2 ratio at 100 degrees for 10 min. Total proteins were subjected to SDS-polyacrylamide gel electrophoresis and immunoblot analysis using anti-CGMMV-CP horseradish peroxidase-conjugated antibodies (CWBIO, Nanjing, China) at a 1:1,000 dilution. Immunoblot signals (18–36 h) were quantified via densitometric analysis using the Image J program (Version 1.8.0, National Institutes of Health, Bethesda, MD, USA) to calculate the silencing efficiency.

## Figures and Tables

**Figure 1 ijms-22-12237-f001:**
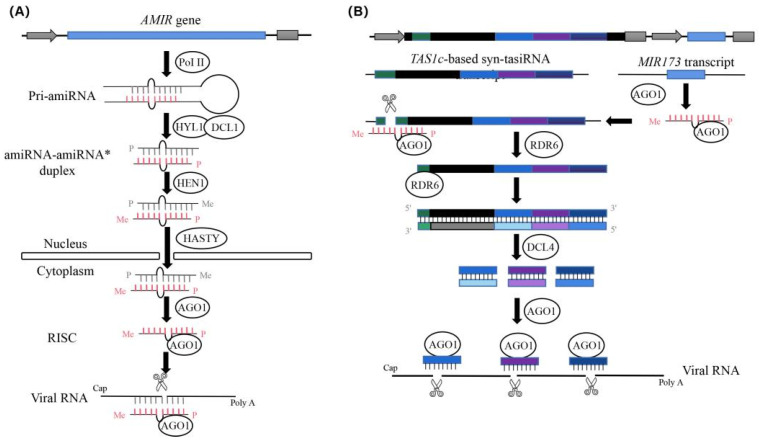
The diagram of the process of antivirus artificial microRNA (amiRNA) and synthetic *trans*-acting small interfering RNA (syn-tasiRNA) production (revised from Jones-Rhoades et al. and Carbonell et al.). (**A**) Diagram of the biogenesis and activities of artificial microRNA (amiRNA) products. Pri-amiRNAs are transcribed by RNA polymerase II(Pol II). The amiRNA-amiRNA* duplex produced by DCL1 is then exported to the cytoplasm, possibly through the action of the plant exportin 5 ortholog HASTY. The guide amiRNA strand is then incorporated into AGO proteins to carry out the silencing reactions. (**B**) Diagram of the biogenesis and activities of synthetic *trans*-acting small interfering RNA (syn-tasiRNA) products. Three different antivirus syn-tasiRNAs (in light or dark blue or purple boxes) were inserted into the *TAS1c* gene from *Arabidopsis thaliana* (in black). The miR173 target site (TS) is shown with a green square box. A cassette, including the *A. thaliana* MIR173 precursor (in blue) to generate miR173, was inserted downstream of the syn-tasiRNA cassette. Specific cleavage sites in target sites located in viral RNAs are indicated with black arrows, with TS coordinates indicated in brackets.

**Figure 2 ijms-22-12237-f002:**
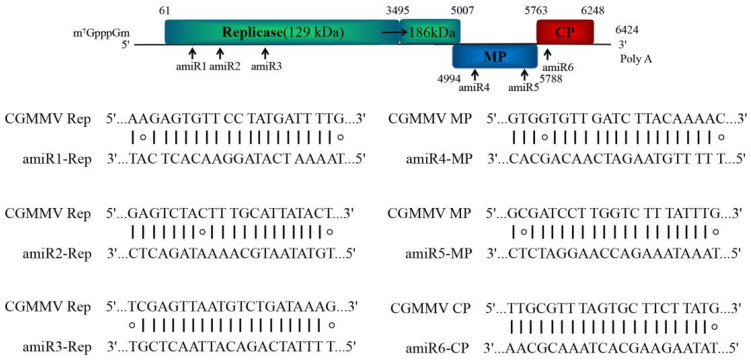
Schematic diagram of amiRNA designed for silencing the CGMMV genes coding for replicase proteins, movement protein (MP), and coat protein (CP). amiRNA, artificial microRNA; CGMMV, cucumber green mottle mosaic virus.

**Figure 3 ijms-22-12237-f003:**
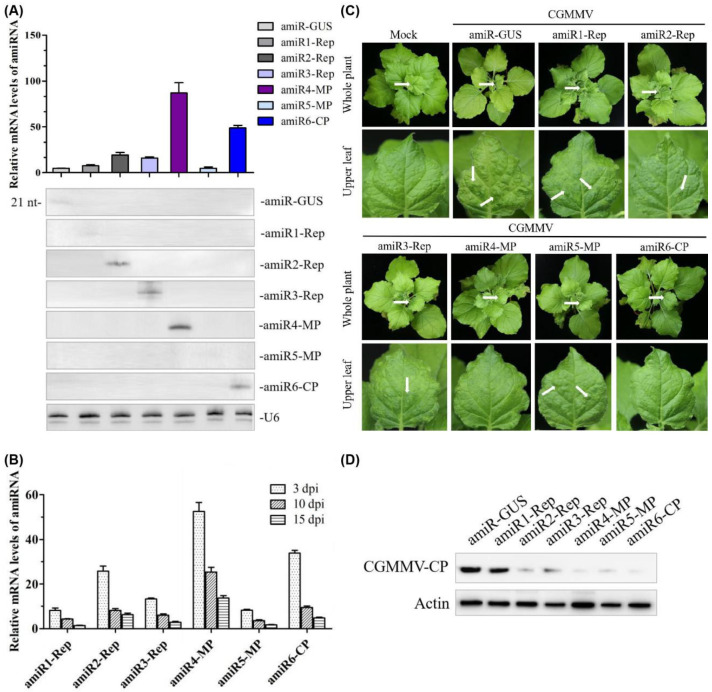
Functional analysis of the anti-CGMMV amiRNA in *Nicotiana benthamiana*. (**A**) amiRNA expression level detected by qRT-PCR (upper panel) and Northern blot hybridization (lower panel); from left to right in upper and lower panels: amiR-GUS, amiR1-Rep, amiR2-Rep, amiR3-Rep, amiR4-MP, amiR5-MP, and amiR6-CP. RNA isolated from agroinfiltrated leaves at 3 dpi. Each biological replicate represents a pool of two agroinfiltrated leaves from the same plant. U6 hybridization serves as a Northern blot loading control. (**B**) Relative expression of amiR1–amiR6 at 3, 10, and 15 dpi in CGMMV-infected *N. benthamiana* plants. Error bars represent the mean of three biological replicates ± standard deviation (SD). (**C**) Photographs of CGMMV-infected whole plants and upper leaves at 20 dpi. Characteristic symptoms of CGMMV-induced mild mottle and mosaic are indicated with white arrows shown in upper leaves, as shown below. White arrows on whole plants indicate the upper leaves. (**D**) Western blot hybridization detection of CGMMV coat protein (17.4 kDa) accumulation in infected upper leaves at 20 dpi. dpi, days post-inoculation. amiR1-Rep, amiR2-Rep, and amiR3-Rep mean the amiRNA targeted three different regions of the Rep gene of CGMMV, respectively; amiR4-MP and amiR5-MP mean the amiRNA targeted two different regions of the MP gene of CGMMV, respectively; amiR6-CP means the amiRNA targeted the CP gene of CGMMV; amiR-GUS means the amiRNA targeted the *GUS* gene.

**Figure 4 ijms-22-12237-f004:**
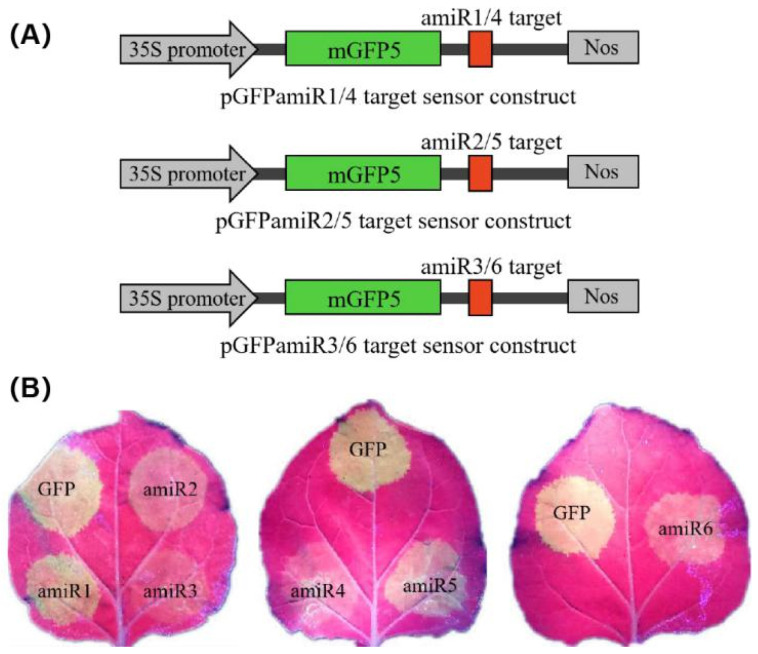
amiRNA functional assays for detecting transient co-expression of amiRNAs and amiR-GFP sensors. (**A**) Schematic representation of three pGFPamiR target sensor constructs, each containing two (amiR1 and amiR4, amiR2 and amiR5, amiR3 and amiR6) amiRNA target sites in the 3′ untranslated region of the mGFP5 reporter gene, under the control of the cauliflower mosaic virus 35S promoter. (**B**) Biological activity of amiRNAs in transient co-infiltration assays. The pGFPamiR target sensor constructs were co-infiltrated with the corresponding amiRNA constructs into *Nicotiana benthamiana* leaves. As a control, the original pEG100-miR171 and either of the pGFPamiR sensor constructs were co-infiltrated. The reduced GFP fluorescence indicates target cleavage by the biological activity of the individual amiRNAs, whereas bright fluorescence indicates the disabled function of the amiRNAs. The three biological repeats of the transient assays showed similar results. GFP, green fluorescent protein.

**Figure 5 ijms-22-12237-f005:**
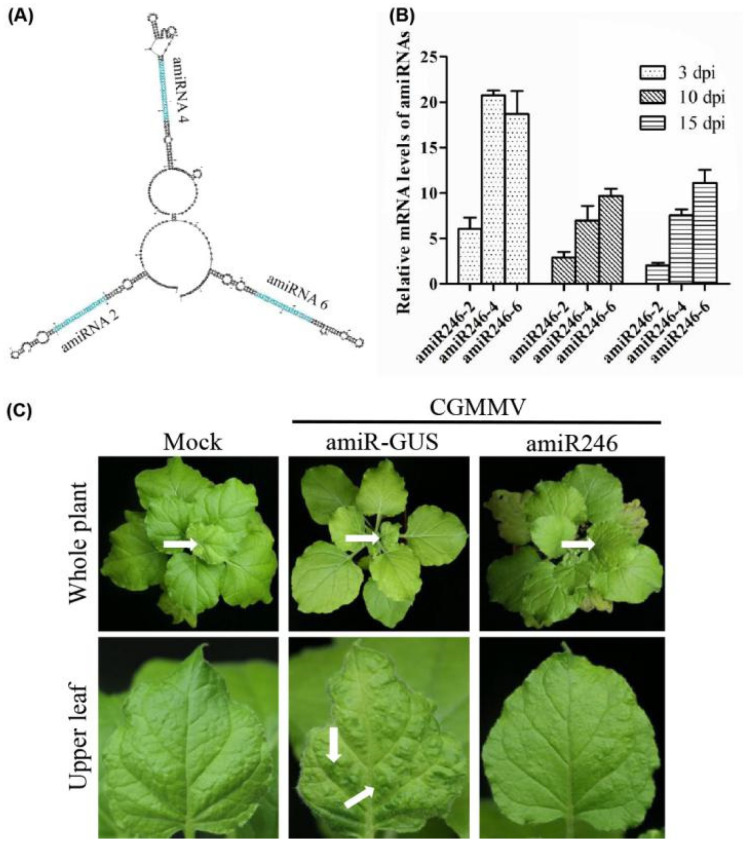
Biological activity of the polycistronic amiR246 to CGMMV in *Nicotiana benthamiana* transient assays. (**A**) Predicted secondary structure of polycistronic amiRNA (amiR246) via mFold analysis. (**B**) Expression levels of three amiRNAs in the polycistronic structure at 3, 10, and 15 dpi after inoculating with CGMMV in *N. benthamiana* plants. For each amiRNA, three biological replicates were evaluated using quantitative reverse transcription-polymerase chain reaction, and three technical replicates were performed for each sample. Error bars represent the mean of three biological replicates ± standard deviation (SD). (**C**) Photographs of the upper leaves and whole plants were taken at 20 dpi. Characteristic symptoms of CGMMV-induced mild leaf mottle and mosaic are indicated with white arrows. White arrows on whole plants indicate the upper leaves.

**Figure 6 ijms-22-12237-f006:**
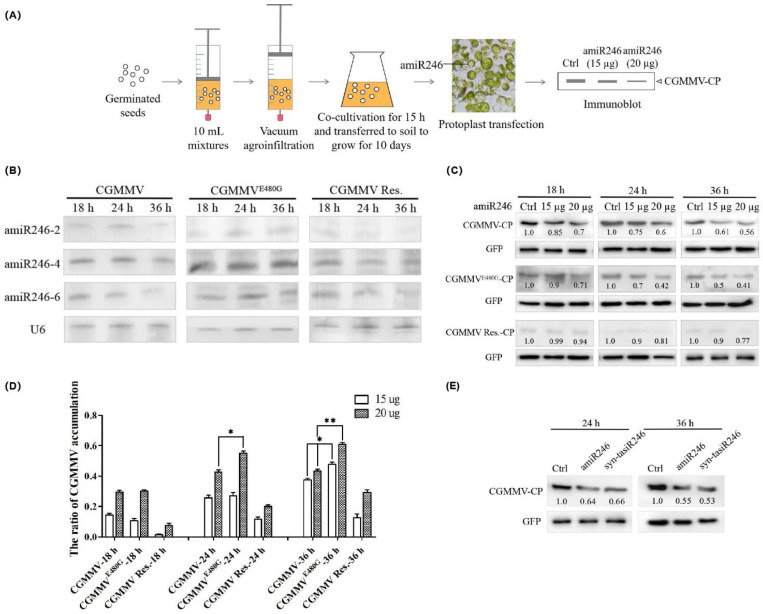
Biological activity of the polycistronic amiR246 construct in CGMMV-infected cucumber protoplasts. (**A**) Schematic diagram of infecting seeds with CGMMV, *Agrobacterium* introduction of amiR246, and protoplast preparation from germinating cucumber seeds. After co-cultivation, the CP protein of CGMMV was detected via immunoassays. (**B**) Northern blot analysis of the expression level of amiRNA at three time points (18, 24, and 36 h) of incubation under different virus strain infections. The U6 blot represents a loading control. (**C**) CGMMV CP levels detected via Western blot in cucumber protoplasts infected with viruses at different incubation time points. Each treatment was repeated in three independent replicates, and the GFP was used as an untargeted internal control. (**D**) Calculation of the inhibition rate of virus accumulation in response to different amounts of amiRNAs (the difference in virus accumulation between the amiRNA group and the control group compared with the control group) shows the inhibitory effect of amiRNAs on virus accumulation of different virulence. Values are the mean ± standard deviation obtained from at least three independent experiments. **, *p*-value ≤ 0.01; *, *p*-value ≤ 0.05. (**E**) Compare the effect of amiR246 and syn-tasiR246 on virus accumulation at two time points (24 and 36 h). CGMMV^E480G^, the mutation of the 480th amino acid of the CGMMV infectious clone vector from glutamic acid (E) to glycine (G). CGMMV Res., the mutation of the target sequences of amiRNAs in the CGMMV infectious clone vector, with less than five mismatches.

**Figure 7 ijms-22-12237-f007:**
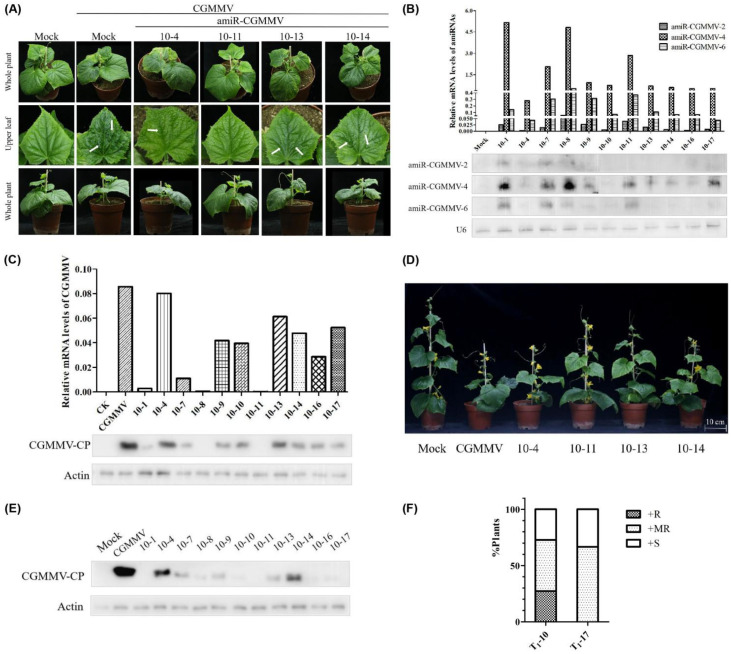
Assessment of CGMMV resistance in transgenic cucumber lines expressed polycistronic amiR246. (**A**) Symptoms on transgenic cucumber plants at 20 dpi; CGMMV-induced typical leaf mild mottle and mosaic are indicated by white arrows. (**B**) Expression of amiR246 in different transgenic cucumber lines detected by Northern blot and qRT-PCR. (**C**) Detection of CGMMV level in transgenic lines by specific qRT-PCR assays (upper) and Western blot hybridization (lower) at 20 dpi. (**D**) Phenotypes of transgenic and control plants after CGMMV infection at 40 dpi. (**E**) Detection of CGMMV accumulation in transgenic plants by Western blot at 40 dpi. (**F**) Western-blot-based bioassay analysis of resistance in segregating populations. +S indicates polycistronic amiR246 transgene-carrying susceptible segregants, +MR indicates transgene-carrying moderately resistant segregants, and +R indicates transgene-carrying resistant segregants. qRT-PCR, quantitative reverse transcription-polymerase chain reaction. amiR-CGMMV, the positive transgenic lines containing the polycistronic amiRNA construct.

## Data Availability

All the data generated or analyzed during this study were included in this article and its Appendix A.

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
