# Peer review of "Polycistronic Artificial microRNA-Mediated Resistance to Cucumber Green Mottle Mosaic Virus in Cucumber"

_ijms, 2021, doi:10.3390/ijms222212237_

Round 1
Reviewer 1 Report
Miao et al. presented a well-crafted work suitable for IJMS. The article is interesting and should be accepted for publication after a minor revision.
Lines 77-85 (suggestion) an illustration/figure of the process of sRNA production will be beneficial for the readability of the introduction.
Lines 104-117 this part is written kind of like an abstract. Please revise by removing the results from the introduction part and focusing on illustrating the objectives of the research.
Lines 192-197, 245-246, 300-305 These parts belong in the discussion part.
Line 415 (and throughout the manuscript) please reference the corresponding figures
Line 426 include the reference
Line 462 revise
Author Response
Point 1: Miao et al. presented a well-crafted work suitable for IJMS. The article is interesting and should be accepted for publication after a minor revision.
Lines 77-85 (suggestion) an illustration/figure of the process of sRNA production will be beneficial for the readability of the introduction.
Response 1: Thanks for your suggestion. A new figure 1 and some information were added to illustrate the process of sRNA production in this paragraph. (Lines 79-88) (Lines 109-120)
Point 2: Lines 104-117 this part is written kind of like an abstract. Please revise by removing the results from the introduction part and focusing on illustrating the objectives of the research.
Response 2: Revised as suggested. I have added some information about the objectives of the research and removed the results from the introduction part. (Lines 121-129)
Point 3: 192-197, 245-246, 300-305 These parts belong in the discussion part.
Response 3: Thanks for your suggestion. I have moved these parts into the discussion.
The sentences from 192-197 have been deleted from this section. Actually, there were similar description in the section of Discussion ‘In this study, amiR2-Rep, amiR4-MP, and amiR6-CP, which were relatively effective in silencing viral RNA, also showed high expression, whereas amiR1-Rep which almost undetectable in northern blot, did not induce resistance to CGMMV (Figure 3).’ (Lines 424-427)
The sentence from 245-246 has been deleted from this section and rewritten as ‘Furthermore, the silencing effect could be enhanced by constructing polycistronic structure targeting multiple sites.’in discussion part. (Lines 435-436)
The sentences from 300-305 have been deleted from this section. There were similar description in the section of Discussion ‘We determined that the mutation of 480th amino acid of CGMMV, which affected RNA silencing suppressor activity, can influence the antiviral effect of amiRNAs by inhibiting the expression of 21 nucleotide amiRNAs (Figure 6B). Moreover, when the dose of amiRNAs reaches a certain level, amiRNAs can overcome the antagonistic effect of the silencing suppressor and induce resistance to the virus (Figure 6C, D).’ (Lines 451-455)
Point 4: Line 415 (and throughout the manuscript) please reference the corresponding figures
Response 4: Revised as suggested. I added “(Figure 6B,C)” at the end of the sentence (Line 412), the figures were cited in corresponding description in Line 427, Line 453, Line 455, Line 467 and Line 475.
Point 5: Line 426 include the reference
Response 5: Revised as suggested. I added “[20]” at the end of the sentence. (Line 424)
Point 6: Line 462 revise
Response 6: This sentence has been revised as the sentence with ‘Notably, similar to siRNA [43], several amiRNAs mediate potential gene silencing via translational repression and/or mRNA cleavage, leading to the decrease of virus accumulation.’ (Lines 458-460)
Reviewer 2 Report
This manuscript reports on research that expands previous works by the same authors, aiming at resistance against cucumber green mottle mosaic virus. The conclusions are supported by the results and the methods used to perform the experiments are correct and well done. Both the introduction and the discussion are very well.
Author Response
Point 1: This manuscript reports on research that expands previous works by the same authors, aiming at resistance against cucumber green mottle mosaic virus. The conclusions are supported by the results and the methods used to perform the experiments are correct and well done. Both the introduction and the discussion are very well.
Response 1: Thanks for your positive comments.
Reviewer 3 Report
RNA silencing is an effective defense system against foreign genetic elements including the cucumber green mottle mosaic virus. This antiviral mechanism has been adopted to generate virus-resistant cultivars through the expression of double-stranded RNA or artificial hairpin RNAs. Miao et al. tested several potential targets for the artificial microRNA and designed a polycistronic artificial microRNA to silence the CGMMV in N. benthamiana and cucumber. Generally, the manuscript is easy to follow and well organized. I have a few suggestions.
Major concerns:
“Polycistronic artificial microRNA-mediated resistance to cucumber green mottle mosaic virus” may be more suitable for the title.
Viruses encode viral suppressors of RNA silencing (VSRs) that affect the RNA silencing efficiency significantly. The VSR region of CGMMV has been identified, somehow. Why do the authors not choose the corresponding regions, but choose the six potential targets tested?
In Figure 4c, does the transient expression of amiR246 trigger programmed cell death in the inoculated leaves? In the mock and amiR-GUS, phenotypes were not observed. Explanation needed.
Fig.7 should be in the supporting data.
Minor:
Figure 2 legend. “N. benthamiana” should be italic.
Line 324, no difference “in” suppression.
Round 2
Reviewer 3 Report
My questions were addressed adequately.